# Post-Thaw Parameters of Buck Semen Quality after Soy Lecithin Extender Supplementation with Fumaric Acid

**DOI:** 10.3390/vetsci10090569

**Published:** 2023-09-12

**Authors:** Aikaterini Saratsi, Foteini Samartzi, Ioannis Panagiotidis, Athina Basioura, Dimitrios Tsiokos, Christina Ligda, Constantinos A. Rekkas

**Affiliations:** 1Veterinary Research Institute, Hellenic Agricultural Organization—DIMITRA, ELGO Campus, 57001 Thermi-Thessaloniki, Greece; ksaratsi@vri.gr (A.S.); samartzi@vri.gr (F.S.); chligda@elgo.gr (C.L.); 2Department of Animal Reproduction & Artificial Insemination, Directorate of Veterinary Center of Thessaloniki, Ministry of Rural Development and Food, 9 Verias Str., 57008 Thessaloniki, Greece; giannis.panagiotidis@gmail.com; 3Department of Agriculture, University of Western Macedonia, Terma Kontopoulou, 53100 Florina, Greece; abasioura@uowm.gr; 4Research Institute of Animal Science, Hellenic Agricultural Organization—DIMITRA, 58100 Paralimni Giannitsa, Greece; tsiokosd@gmail.com

**Keywords:** goat, buck, semen, spermatozoa, cryopreservation, soy lecithin extender, fumaric acid

## Abstract

**Simple Summary:**

The cryopreservation of spermatozoa causes biochemical and ultrastructural changes that compromise their structural and functional integrity, thereby affecting their fertilizing ability. These alterations are influenced and/or mediated by oxidative stress. Since goat seminal plasma interacts with egg yolk and milk constituents, soy-lecithin-based extenders have been used for the cryopreservation of goat spermatozoa for over a decade. Fumaric acid, a substance that enhances antioxidant enzyme activity, has been extensively used in therapeutics due to its antioxidant, anti-inflammatory and immunoregulating properties. We tested the addition of fumaric acid (0 mM, 2.15 mM, 10 mM, 30 mM) in a soy lecithin extender on post-thaw buck sperm quality and function variables (sperm motility and kinetics (CASA); viability (eosin–nigrosin); acrosome integrity (SpermBlue^®^); membrane functional integrity (hypo-osmotic swelling test; HOST) and mitochondrial function (MF; Rhodamine 123/SYBR-14/PI)). Our results indicated a beneficial effect of 2.15 mM fumaric acid supplementation on frozen–thawed buck sperm viability, acrosome integrity, plasma membrane functional integrity and mitochondrial function. Further research related to sperm-fertilizing capacity under in vivo conditions is currently in progress.

**Abstract:**

The supplementation of cryopreservation media with antioxidants improves the post-thaw quality and fertilizing ability of spermatozoa. To maximize the fertility of frozen–thawed buck spermatozoa, further research is required to overcome obstacles that have yielded controversial results and standardize protocols. In the present work, the effect of adding fumaric acid (a well-described antioxidant) to a soy lecithin semen extender on certain quality parameters of spermatozoa following freezing and thawing was examined for the first time. Five sexually mature Skopelos bucks were used, and ejaculates were collected with an artificial vagina. The semen samples (98 samples, five replicates) were diluted (400 × 10^6^ spermatozoa/mL) with OviXcell^®^, supplemented with fumaric acid (0 mM, 2.15 mM, 10 mM or 30 mM), equilibrated (5 °C; 3 h), packed (0.5 mL straws), frozen and stored (−196 °C) until further processing. After thawing, the spermatozoa total and progressive motility (CASA), viability (eosin–nigrosin), membrane functional integrity (HOST), acrosome integrity (SpermBlue^®^) and mitochondrial function (Rhodamine-123/SYBR-14/PI) were evaluated. Statistical analysis was performed with one-way ANOVA, followed by Duncan’s test; significance was set at 0.05. The addition of 2.15 mM fumaric acid improved (*p* < 0.05) spermatozoa viability, membrane functional integrity, acrosome integrity and mitochondrial function compared to all other concentrations. The addition of 30 mM fumaric acid decreased (*p* < 0.05) spermatozoa viability and mitochondrial function compared to all other concentrations. These results indicate a beneficial effect of a 2.15 mM fumaric acid addition to a soy lecithin extender on post-thaw buck spermatozoa quality. Further research is required to evaluate the in vivo fertility of frozen–thawed buck spermatozoa treated with fumaric acid, as well as to elucidate the mechanism of action of fumaric acid in spermatozoa.

## 1. Introduction

Artificial insemination (AI) with cryopreserved semen is one of the main assisted reproduction techniques used in goat breeding programs. Cryopreserved semen is essential to the development of gene banks for endangered breeds and the dissemination of valuable genetic material [1,2]. However, cryopreservation decreases the post-thaw motility and viability of buck spermatozoa, reduces their fertilizing ability and, finally, limits the extensive use of frozen–thawed buck semen under field conditions [2,3,4,5].

Oxidative stress, one of the crucial contributing factors to spermatozoa cryo-injuries, is a result of excessive intracellular reactive oxygen species (ROS) formation [6]. The main organelles of ROS formation in spermatozoa are the mitochondria [7]. During the freeze–thaw procedure, spermatozoa are exposed to cold shock and air (O_2_), and increased ROS formation downregulates the levels of endogenous antioxidants [8].

ROS production is a normal process for spermatozoa, as they play pivotal roles in sperm motility, capacitation, acrosome reaction, sperm-oocyte interactions and fertilization [9,10,11]. ROS have beneficial or detrimental effects on sperm functions depending on their nature, their concentration and the duration of spermatozoa exposure [9,12]. A set of enzymatic scavenging systems against free radicals regulates ROS levels in the mitochondria and the cytoplasm [13]. Endogenous antioxidants prevent excessive ROS formation and downregulate the activation of the apoptotic cascade [14,15]. They either directly scavenge free radicals or use antioxidant signaling pathways and act mostly through receptors to enhance antioxidant activity [16].

Excessive ROS production overwhelms the natural antioxidant capacity of spermatozoa, thereby inducing membrane lipid peroxidation. The redox imbalance gradually causes the disruption of mitochondrial functions and DNA damage in spermatozoa and results in post-thaw motility suppression [17,18]. The progressing redox imbalance finally compromises membrane integrity and spermatozoa viability [19].

Cryoprotective supplements for freezing extenders possess lipid properties that directly combat ROS [20,21]. Furthermore, the supplementation of extenders with antioxidants restores the equilibrium between ROS production and scavenging, protects spermatozoa metabolism and prevents cryo-damage, thereby improving the post-thaw quality and fertilizing ability of spermatozoa [8,22,23].

In addition to other difficulties arising in the cryopreservation of small ruminant spermatozoa, buck semen contains the egg yolk coagulating enzyme (EYCE), which interacts with egg yolk, and BUSgp60, which hydrolyses skim-milk extender components, both producing metabolites toxic to spermatozoa [2,24]. Soy lecithin extenders have therefore been used as an alternative to egg yolk in buck semen cryopreservation [25,26].

Fumaric acid (FA) is a naturally occurring organic acid omnipresent in living organisms. Chemically, it is an unsaturated dicarboxylic acid (C_4_H_4_O_4_), first isolated from Fumaria officinalis [27]. FA is a carboxylic acid of the Krebs cycle, and, as such, it is normally found in mammalian semen [28,29]. FA and fumaric acid esters (FAEs) are well-known antioxidants. They have strong ROS scavenging properties and exhibit anti-inflammatory, immunomodulatory and chemopreventive effects on cells, tissues and organs [30,31]. FA and FAEs affect the glutathione redox cycle, thereby exerting their antioxidant effects [32]. To regulate the gene transcription of antioxidant enzymes, FAEs use the nuclear factor (erythroid-derived 2)-related factor 2 (NRF2) genetic pathway [33,34], thereby increasing intracellular reduced glutathione (GSH) and regulating cytokine formation, cell proliferation and apoptosis [35,36,37]. The interplay between fumarates and the glutathione enzymatic system depends upon cell type, fumarate type and metabolite and concentration [38,39].

Dimethyl fumarate (DMF), one of the major FAEs, administered as a dietary supplement to rats, exerts gene-regulated anti-apoptotic effects in Leydig cells, increases serum testosterone concentrations and influences sperm maturation and acrosome formation [40]. 

The in vitro effect of FAEs on spermatozoa has only been evaluated in the boar. Magnesium fumarate, used as an antioxidant supplement in the liquid storage of boar semen (15 °C), increases sperm survival, though medium and high concentrations of it result in elevated percentages of spermatozoa chromatin damage [41].

In the present study, we chose Skopelos-breed bucks, an important local goat breed, included in an established breeding program in Greece to examine the effects of three distinct concentrations of fumaric acid addition to a soy lecithin extender on post-thaw spermatozoa quality and function variables in vitro. We chose fumaric acid in light of its antioxidant activity on various cell lines and organs. The supplementation of freezing extenders with yet-untested antioxidants may assist in the effort to enhance the percentage of fertile spermatozoa in cryopreserved buck semen. A preliminary analysis of part of the data presented here, [42], has indicated a possible beneficial effect of fumaric acid supplementation to soy-lecithin-based extenders on post-thaw buck sperm quality in vitro.

## 2. Materials and Methods

### 2.1. Selection of Bucks, Semen Collection and Handling

This study was performed at the facilities of the Department of Animal Reproduction & Artificial Insemination, Directorate of Veterinary Center of Thessaloniki, Ministry of Rural Development and Food. All animal procedures were performed in accordance with European Union Regulation 2010/63.

Skopelos bucks were selected on the basis of data collected from the genetic improvement program of the breed, as performed by the Breeders’ Association (Livestock Cooperative of Volos, Skopelos goat breed). The Association has maintained a comprehensive database that encompasses the genealogical and production records of the breeding population. By examining the genealogical information, we selected unrelated animals and used production data from their relatives to objectively evaluate the potential of each candidate buck.

Five Skopelos bucks, 2–3 years old, were housed in individual pens under natural daylight conditions (40″68′ north latitude). They were fed mixed-grass hay and concentrate as needed to maintain a healthy body condition score and had access to fresh water ad libitum.

Semen was collected with an artificial vagina, once a week, from April to June. A maximum of two ejaculates, with a minimum 15 min interval between ejaculations, was collected per animal per collection date and then pooled. The semen samples were placed in a water bath at 37 °C immediately after collection. Initially, semen volume was recorded and spermatozoa concentration was evaluated with a spectrophotometer (DR Lange LP 1 Photometer; Minitüb GmbH, Tiefenbach, Germany). Spermatozoa motility was subjectively evaluated, always by the same person, using a standard dilution (1:200) in PBS (37 °C) under a phase-contrast microscope (×400). At least five random fields were observed for each sample. Eosin–nigrosin staining was used for the evaluation of spermatozoa viability and morphology (200 spermatozoa per slide were examined). The ejaculates that provided >70% live and motile spermatozoa and <5% morphological abnormalities were further processed [43]. Five semen samples were collected from each male, and a total of 98 samples were used in five replicates.

### 2.2. Semen Extender, Fumaric Acid Addition

Each ejaculate was split into four parts of equal volume and each part diluted to a final concentration of 400 × 10^6^ spermatozoa/mL; OviXcell^®^ (IMV Technologies, Saint Uuen sur Iton, L’Aigle, France), an animal protein-free ram semen extender containing soy lecithin, was used for semen dilution The basic diluent (OviXcell^®^) was supplemented with one of four concentrations (0 mM (control), 2.15 mM, 10 mM or 30 mM) of fumaric acid (Fumaric Acid Ref Nr 8509; Sigma-Aldrich, Steinheim am Albuch, Germany).

Fumaric acid concentrations were chosen, after preliminary testing, based on former studies of boar semen [41] and other in vitro studies of cell cultures of various cell lines, which have demonstrated the cytoprotective, antioxidant and anti-inflammatory effects of fumaric acid and its esters [30,44,45].

### 2.3. Semen Cryopreservation

After equilibration at 5 °C for 3 h, all semen samples were packed into 0.25 mL French straws (400 × 10^6^ spermatozoa/mL) and frozen in two steps, according to Purdy [24], with minor modifications. The straws were kept at 10 and 4 cm over a liquid nitrogen (LN_2_) surface for 3 and 2 min, respectively, then immersed in LN_2_ and finally stored in a LN_2_ tank (−196 °C) for at least one month.

### 2.4. Semen Evaluation

The frozen semen samples were thawed in a water bath at 37 °C for 30 s. All tests were performed immediately after thawing, by the same person.

#### 2.4.1. Spermatozoa Motility

The spermatozoa motility parameters were evaluated with a Computer-Assisted Spermatozoa Analysis (CASA) system (Sperm Class Analyser^®^; Microptic S.L., Automatic Diagnostic Systems, Barcelona, Spain) and a microscope (AXIO Scope A1; Zeiss, Oberkochen, Germany) equipped with a heating stage and a camera (Basler scA780 54fc; Basler AG, Ahrensburg, Germany). This analysis was performed with Sperm Class Analyser^®^ software (SCA^®^ v.6.3.; Microptic S.L., Automatic Diagnostic Systems, Barcelona, Spain). The CASA configurations were (1) at least 5 fields recorded (×100) for each semen sample, (2) >500 spermatozoa, (3) 25 frames/s, (4) a region of particle control of 3–70 microns and (5) progressive movement of >80% of the STR parameter.

For each sample, 10 μL of semen was placed on the preheated (37 °C) Makler chamber (Makler^®^ counting chamber, 10 μm deep; Sefi Medical Instruments, Haifa, Israel) and the following CASA motility parameters were evaluated: (1) total motility (%) and (2) progressive motility (%).

#### 2.4.2. Spermatozoa Viability

Spermatozoa viability was assessed using eosin–nigrosin staining (1% eosin, 3% nigrosin, 3% sodium citrate, 100 mL of distilled water). Fifteen μL of semen and an equal amount of eosin–nigrosin stain were mixed and smeared on each slide with a coverslip. The slides were left to air dry and were evaluated for spermatozoa viability. At least 200 spermatozoa per slide were counted (×400).

#### 2.4.3. Plasma Membrane Functional Integrity

Assessment of the spermatozoa plasma membrane functional integrity was performed with the hypo-osmotic swelling test (HOST), according to Fonseca et al. [46]. At least 200 spermatozoa per slide were counted (×400).

#### 2.4.4. Acrosome Integrity

Acrosome integrity was evaluated, according to Van der Horst and Maree [47], using the SpermBlue^®^ stain kit (Microptic, Automatic Diagnostic Systems, Barcelona, Spain). At least 200 spermatozoa per slide were counted (×400) using a blue filter (Leica DMLB).

#### 2.4.5. Mitochondrial Membrane Function Combined with Viability

Mitochondrial function was evaluated with Rhodamine 123 [48], while a concurrent viability assessment was performed with a SYBR-14/PI assay [48,49]. Samples were exposed to the three dyes in the dark at room temperature (20–25 °C). By the end of a 25 min incubation period, the samples were assessed (×400) under a fluorescent microscope (Leica DM 2000; Leica Microsystems, Balgach, Switzerland, Ltd.).

### 2.5. Statistical Analysis

All data obtained from this study were analyzed using SPSS (SPSS 20.0 for Windows; SPSS, Chicago, IL, USA). A one-way analysis of variance (ANOVA) was applied, followed by Duncan’s multiple range test; the homogeneity of variances was evaluated with the Levene test, and *p* < 0.05 was set as the minimum level of significance. Data are presented as means ± standard deviation (SD).

## 3. Results

### 3.1. CASA Motility

The spermatozoa total and progressive motility (%) were significantly lower in the extender containing 30 mM fumaric acid compared to all other extenders (*p* < 0.0005), while no significant differences (*p* > 0.05) were observed among the extenders containing 0, 2.15 and 10 mM fumaric acid (Figure 1 and Figure 2).

### 3.2. Spermatozoa Viability (Eosin–Nigrosin)

The percentage of live spermatozoa was significantly higher in the 2.15 mM fumaric acid concentration in comparison to all other concentrations (*p* < 0.0005). Furthermore, the percentage of live spermatozoa was significantly lower (*p* < 0.0005) in the 30 mM fumaric acid concentration when compared to all other concentrations (Figure 3).

### 3.3. Plasma Membrane Functional Integrity

The percentage of spermatozoa with functional cell membranes was significantly higher in the 2.15 mM fumaric acid concentration compared to all other concentrations (*p* < 0.0005), while no significant differences (*p* > 0.05) were observed among all the other concentrations (0, 10 and 30 mM) of fumaric acid (Figure 4).

### 3.4. Acrosome Integrity

The percentage of spermatozoa with intact acrosome membranes was significantly higher in the 2.15 mM fumaric acid concentration compared to all other concentrations (*p* < 0.0005). No significant differences (*p* > 0.05) were observed among all the other concentrations (0, 10 and 30 mM) of fumaric acid (Figure 5).

### 3.5. Mitochondrial Membrane Function Combined with Viability

The percentage of live spermatozoa with high mitochondrial membrane potential (MMP) was significantly higher in the 2.15 mM fumaric acid concentration in comparison to all other concentrations (*p* < 0.0005). The percentage of live spermatozoa with high mitochondrial membrane potential was lower (*p* < 0.0005) in the 30 mM FA concentration in comparison to all the other concentrations (Figure 6).

## 4. Discussion

The main results of the present study indicate that the supplementation of a soy lecithin extender with the appropriate concentration of fumaric acid (2.15 mM) has a positive effect on Skopelos buck spermatozoa quality parameters, while at higher concentrations fumaric acid exhibits no effect (10 mM) or a negative effect (30 mM). Specifically, fumaric acid at the concentration of 2.15 mM improved post-thaw buck spermatozoa viability compared to the control (0 mM) and the higher concentrations of fumaric acid (10 or 30 mM). Interestingly, the viability was suppressed by the highest concentration of fumaric acid (30 mM). Fumaric acid at the 2.15 mM concentration also enhanced the plasma and mitochondrial membrane functionality and acrosome integrity. The addition of fumaric acid to the soy lecithin extender at low (2.15 mM) or medium (10 mM) concentrations failed to further enhance total and progressive motility of frozen–thawed buck spermatozoa, while a negative effect was observed with the higher concentration of fumaric acid (30 mM).

Fumaric acid esters (FAEs) regulate mitochondrial function and are associated with the regulation of the glutathione redox cycle [32,35]. Depending on the dose and type of fumarate, FAEs can decrease or enhance cell viability, mitochondrial oxygen consumption and glycolysis rates [30]. This mode of action could also be the underlying cause of the negative effect exerted by the 30 mM concentration of fumaric acid on spermatozoa viability, mitochondrial functionality and motility in the present study. It may be assumed that the highest concentration is likely to cause depletion of GSH with all of the abovementioned deleterious effects not only on sperm mitochondria but also on cell viability and motility. Similarly, glutathione, when added as a supplement in the optimum range of concentrations, increases the post-thaw motility, viability and plasma membrane and acrosomal integrity of buffalo spermatozoa, while higher-than-optimum concentrations exhibit no protective effects [50]. Interestingly, when an even higher range of concentrations was used, glutathione had a deleterious effect on ram spermatozoa [51,52]. The same effect has been reported for the amino acid glutathione precursors of intracellular GSH, proline, betaine glycine, glycine and cysteine on post-thaw motility, viability and membrane integrity of ram spermatozoa [53,54].

A reversal of the in vitro effect of magnesium fumarate on spermatozoa has been observed in the liquid storage of boar semen. Magnesium fumarate, added at a low concentration, increases sperm survival compared to the basic extender, while its medium and high concentrations result in spermatozoa chromatin damage [41].

A similar reversal of action, associated with the concentration of antioxidants in semen freezing extenders, is exerted by α-tocopherol and its water-soluble analogue, Trolox, in various animal species and humans [55,56,57], as well as by mitoquinone or procyanidins in ram freezing extenders [58]. Reduced glutathione, Trolox, crocin and cysteamine, in combination with egg yolk or soy lecithin extenders, also exhibit controversial effects on frozen–thawed ram spermatozoa [59].

Furthermore, our results may have been affected by the presence of soy lecithin, which possesses cryoprotectant properties. Cryoprotectants affect the efficiency of antioxidant supplements by contributing to the total antioxidant activity of the extenders [59,60]. Soy lecithin effectively protects spermatozoa against apoptosis and cell death during the freeze–thaw process when compared to egg yolk [60,61,62,63]. However, in contrast to egg yolk, soy lecithin induces mitochondrial membrane damage, but this is not immediately reflected in the spermatozoa motility. These alterations become more evident as post-thawing incubation time increases and may negatively affect the spermatozoa fertilizing ability [62].

It is possible that the beneficial effect of FA on frozen–thawed buck spermatozoa is mediated through the activation of the NRF2 pathway, as is the case in numerous other cells and tissues [30,36,44,64,65]. NRF2 activation in spermatozoa regulates gene expression of endogenous antioxidants and cytoprotective enzymes by binding directly to the promoters of antioxidant genes [66,67]. Moreover, it has been proven that in human spermatogenesis, NRF2 mRNA expression is associated with ejaculate spermatozoa concentration, morphology, motility and viability, while low levels of NRF2 expression are clearly linked to infertility [67,68].

Another antioxidant supplement of sperm freezing extenders that acts through the NRF2 pathway and seems to act like fumaric acid is melatonin. Melatonin increases NFR2 mRNA levels and its downstream genes [69,70]. It maintains mitochondrial membrane potential (MMP) and preserves mitochondrial function through scavenging ROS, thus enhancing the motility, viability, membrane integrity and mitochondrial activity of post-thaw human spermatozoa [71]. Fumaric acid may be using the same pathway in buck spermatozoa, but that is a matter that requires further research.

As to the protective effect of fumaric acid on MMP, it is yet unclear why in the current study, this protective effect did not result in higher total and progressive motility values of post-thaw buck spermatozoa, since MMP is positively correlated with spermatozoa motility [72]. When MMP is measured with fluorescent stains, the average fluorescence value is correlated with mitochondria oxygen consumption and progressive motility [48,73,74]. However, in vitro cultures of human spermatozoa have proven that a loss of motility on account of oxidative stress can occur before the decrease in MMP, since the cascade progressing toward the loss of mitochondrial function can affect spermatozoa motility in a preliminary stage [75].

Several studies have indicated that the assessment of MMP can predict the fertilizing ability of spermatozoa both in vivo and in vitro [76,77]. In human spermatozoa, high MMP is correlated with acrosin activity and is essential for acrosome reaction induction and hyperactivation and the protection of chromatin integrity [74,78]. Thus, MMP can be a marker for a multitude of orchestrated functions that influence spermatozoa viability and fertilizing ability. We may hypothesize that the protective effect of 2.15 mM fumaric acid on sperm mitochondria is reflected in the enhancements of viability and acrosome integrity. Spermatozoa motility and fertilizing ability are closely correlated with mitochondrial function, while mitochondria functionality has been indicated as a biomarker of sperm quality in several species [79,80,81].

The negative effect of the 30 mM fumaric acid concentration on the MMP of buck frozen–thawed spermatozoa not only is mirrored in spermatozoa viability but also results in a significant drop in total and progressive post-thaw motility. Exogenous antioxidants should be used with caution, as studies have shown that their beneficial effects are concentration-dependent or can be reversed [50,56,59].

Fumaric acid, in the concentration range we applied, has a well-established role as an antioxidant in both pure systems and cell cultures [30,31,32]. Thus, it is highly probable that it exerts its protective effect through the alleviation of oxidative stress and the restoration of the redox balance in frozen–thawed buck spermatozoa. However, further in vitro research is required concerning how the addition of fumaric acid influences the degrees of lipid peroxidation and DNA fragmentation, total antioxidant capacity (TAC) and glutathione peroxidase and superoxide dismutase activity before the underlying mechanisms of action of fumaric acid can be clarified. The possible genetic program used by fumaric acid, i.e., the activation of the NRF-2 pathway, is also a subject for further investigation. Its confirmation as the mode of action of fumaric acid on goat spermatozoa will not be a surprise as far as the acid is concerned but would provide insight into the physiology of goat spermatozoa. Moreover, since it is well-known that semen quality parameters in vitro only indicate the potential of the in vivo fertilizing capacity, to assess the practical value of these findings for the enhancement of frozen buck semen fertilityin the context of artificial insemination, in vivo fertility trials are in order, since the beneficial effects on in vitro parameters need to be confirmed with an upgraded pregnancy rate to constitute an improvement in reproductive success.

## 5. Conclusions

In conclusion, our results indicate that fumaric acid, added to a soy lecithin extender at a 2.15 mM concentration, exerts a beneficial effect on post-thaw buck spermatozoa viability, plasma and acrosome membrane integrity and mitochondrial function. Further research considering (1) the mechanism(s) of fumaric acid’s beneficial and detrimental effects on buck spermatozoa during freezing/thawing and (2) the impact of fumaric acid’s addition to buck semen extenders on the actual fertilizing capacity of buck spermatozoa following freezing/thawing is necessary to establish a role for fumaric acid use in buck semen processing.

## Figures and Tables

**Figure 1 vetsci-10-00569-f001:**
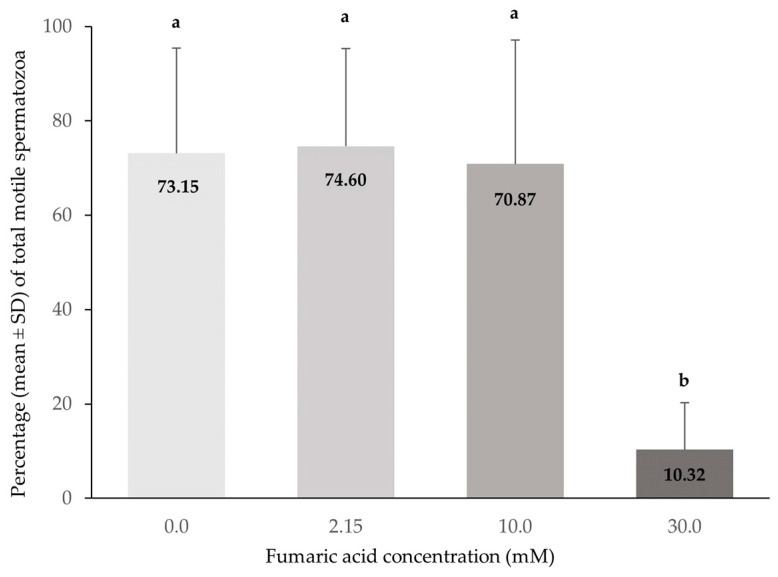
Percentages (means ± SD) of total motile spermatozoa (CASA) in Skopelos buck semen cryopreserved in commercial soy-lecithin-based extender (OviΧcell^®^) and supplemented with 0 mM (*n* = 25), 2.15 mM (*n* = 24), 10 mM (*n* = 25) and 30 mM (*n* = 24) fumaric acid. a, b: Bars with different letters differ significantly (*p* < 0.0005).

**Figure 2 vetsci-10-00569-f002:**
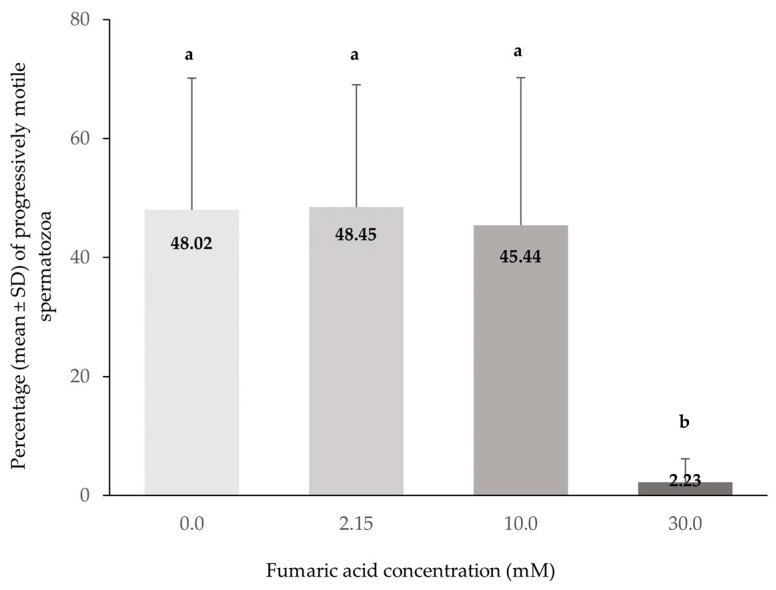
Percentages (means ± SD) of progressively motile spermatozoa (CASA) in Skopelos buck semen cryopreserved in commercial soy-lecithin-based extender (OviΧcell^®^) and supplemented with 0 mM (*n* = 25), 2.15 mM (*n* = 24), 10 mM (*n* = 25) and 30 mM (*n* = 24) fumaric acid. a, b: Bars with different letters differ significantly (*p* < 0.0005).

**Figure 3 vetsci-10-00569-f003:**
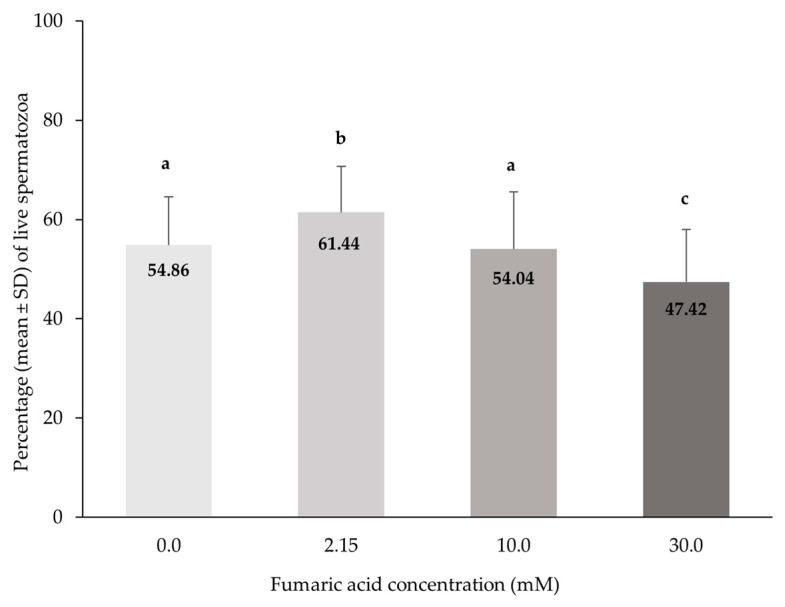
Percentages (means ± SD) of live spermatozoa (eosin–nigrosin) in Skopelos buck semen cryopreserved in commercial soy-lecithin-based extender (OviΧcell^®^) and supplemented with 0.0 mM (*n* = 25), 2.15 mM (*n* = 24), 10.0 mM (*n* = 25) and 30.0 mM (*n* = 24) fumaric acid. a, b, c: Bars with different letters differ significantly (*p* < 0.0005).

**Figure 4 vetsci-10-00569-f004:**
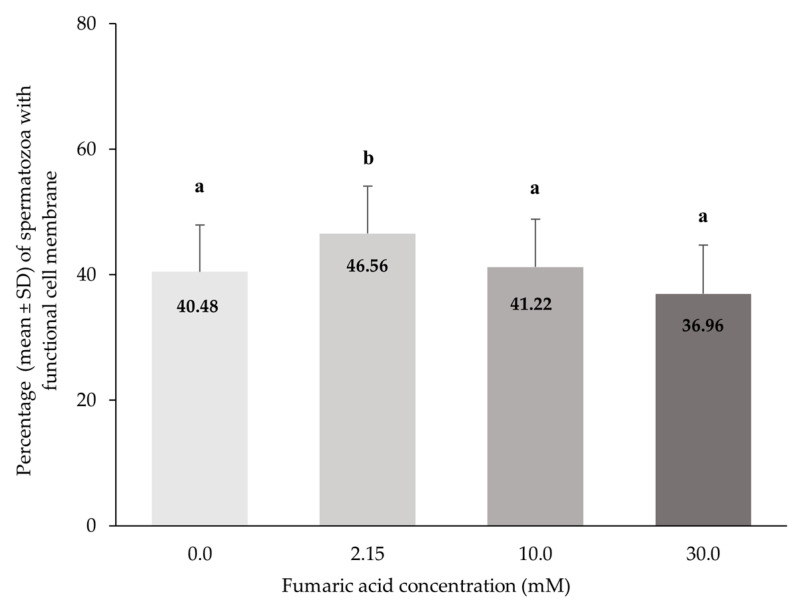
Percentages (means ± SD) of spermatozoa with functional cell membranes (HOST) in Skopelos buck semen cryopreserved in commercial soy-lecithin-based extender (OviΧcell^®^) and supplemented with 0.0 mM (*n* = 25), 2.15 mM (*n* = 24), 10.0 mM (*n* = 25) and 30.0 mM (*n* = 24) fumaric acid. a, b: Bars with different letters differ significantly (*p* < 0.0005).

**Figure 5 vetsci-10-00569-f005:**
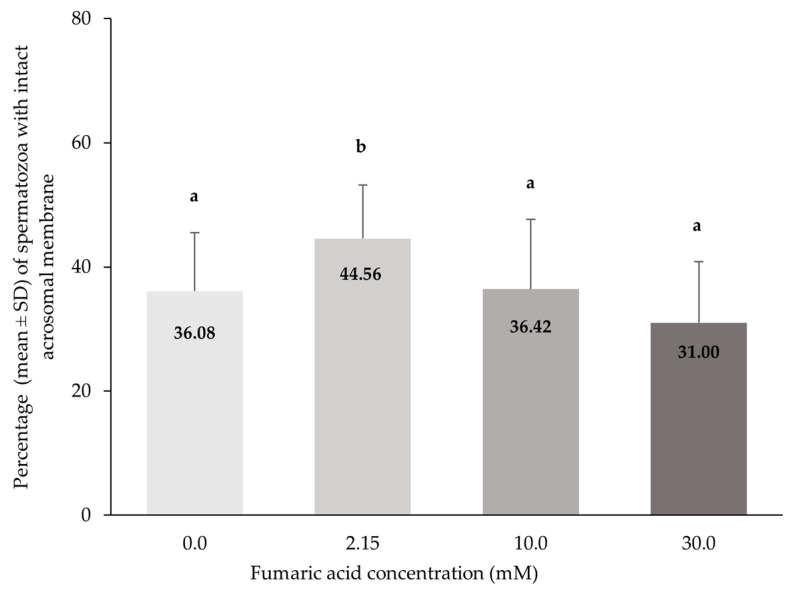
Percentages (means ± SD) of spermatozoa with intact acrosomal membranes (SpermBlue^®^) in Skopelos buck semen cryopreserved in commercial soy-lecithin-based extender (OviΧcell^®^) and supplemented with 0.0 mM (*n* = 25), 2.15 mM (*n* = 24), 10.0 mM (*n* = 25) and 30.0 mM (*n* = 24) fumaric acid. a, b: Bars with different letters differ significantly (*p* < 0.0005).

**Figure 6 vetsci-10-00569-f006:**
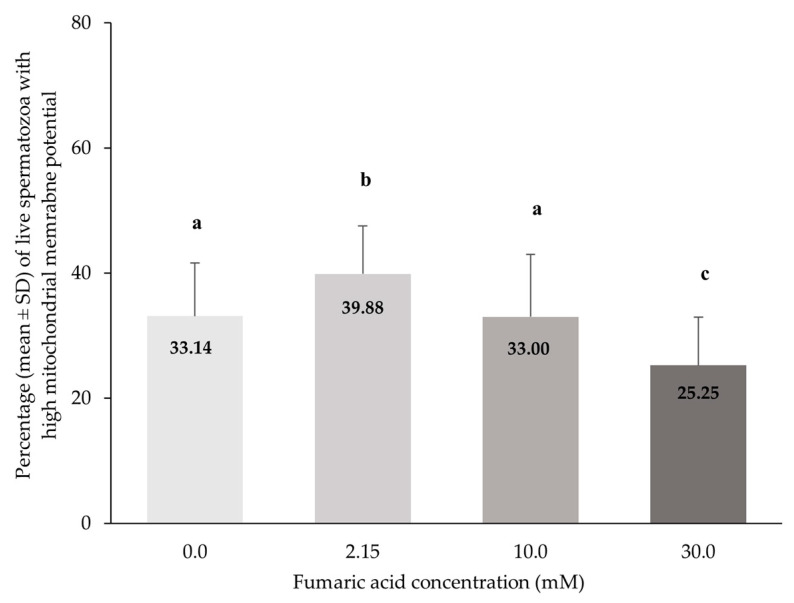
Percentages (means ± SD) of live spermatozoa with high mitochondrial membrane potential (SYBR-14/PI) in Skopelos buck semen cryopreserved in commercial soy-lecithin-based extender (OviΧcell^®^) and supplemented with 0.0 mM (*n* = 25), 2.15 mM (*n* = 24), 10.0 mM (*n* = 25) and 30.0 mM (*n* = 24) fumaric acid. a, b c: Bars with different letters differ significantly (*p* < 0.0005).

## Data Availability

Data supporting the reported results are available to anyone interested after justified application is provided.

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
