# Peer review of "Post-Thaw Parameters of Buck Semen Quality after Soy Lecithin Extender Supplementation with Fumaric Acid"

_vetsci, 2023, doi:10.3390/vetsci10090569_

Round 1
Reviewer 1 Report
Overall, the manuscript provides a clear and concise overview of the research conducted on the effect of fumaric acid supplementation in a soy lecithin extender on frozen-thawed buck semen quality. However, there are a few minor points to consider for improvement:
In the Simple Summary, Lines 13-14:
The phrase "effects can be attributed to some extent, to oxidative stress" could be clarified by mentioning that oxidative stress is known to affect spermatozoa's structural and functional changes during cryopreservation.
Lines 14-15:
“Goat spermatozoa are more susceptible to oxidative stress, compared to those of other mammalian species, as they possess a lower percentage of membrane phospholipids and a higher ratio of polyunsaturated fatty acids”.
While this statement provides essential insight into the justification of your hypothesis, it lacks sufficient elaboration and referencing. Provide a more detailed explanation in the Discussion section or the Introduction.
Please consider elaborating on why goat spermatozoa exhibit higher susceptibility to oxidative stress due to their lower percentage of membrane phospholipids and higher ratio of polyunsaturated fatty acids. Support the above statement with references.
In both the Simple Summary and Abstract, consider providing a bit more context about the significance of the study. Why is it important to improve post-thaw sperm quality? Mention any potential applications or benefits of the findings.
In the Abstract and Materials and Methods, specify the total number of replicates or experiments performed for each treatment group. The authors do not explicitly state the sample size used for the statistical analysis. Including this information would be helpful to give readers a better understanding of the study's robustness.
Consider adding a sentence to the Abstract to highlight that the addition of fumaric acid to the soy lecithin extender is novel or unique in the context of cryopreservation of buck spermatozoa.
In the Abstract, when mentioning the statistical analysis, specify the significance level (e.g., p < 0.05) used to determine statistical significance.
The "Introduction" section provides a clear overview of the research context and rationale for the study.
The "Materials and Methods" section provides a well-structured and complete description of the experimental procedures used in the study. However, it appears that the authors did not explicitly mention the total number of semen samples used in the investigation. It would be beneficial for readers to have this essential information to understand the statistical power of the study and the validity of the results. I recommend that the authors include the total number of semen samples collected from each male, as well as the number of replicates or repeats performed for each treatment or fumaric acid concentration. This clarification is necessary to ensure the robustness of the study results.
The "Discussion" section is well-organized and coherent, providing a thorough analysis of the study's findings. The authors have justified their results based on the literature and have made appropriate connections between their results and the effects of fumaric acid on various parameters of spermatozoa quality. The section is also well-referenced, supporting their claims with relevant studies.
The discussion adequately addresses the positive effects of fumaric acid at the concentration of 2.15mM on Skopelos buck spermatozoa quality parameters, including viability, plasma and mitochondrial membrane functionality, and acrosome integrity. It also highlights the negative effect of the higher concentration (30mM) of fumaric acid on spermatozoa viability, motility, and mitochondrial membrane potential. The authors have presented plausible explanations for these effects, attributing them to the regulation of the glutathione redox cycle, mitochondrial function, and the activation of the NRF2 pathway.
In terms of the discussion's coherence, it flows logically from one point to another, and the authors have addressed potential mechanisms underlying the observed effects. The discussion also raises interesting points about the concentration-dependent effects of antioxidants in sperm freezing extenders and the potential role of soy lecithin in the study's outcomes.
However, one aspect that could be further elaborated upon is the potential implications of the study's findings. The authors could discuss how the improved post-thaw buck spermatozoa quality parameters, observed at the 2.15mM fumaric acid concentration, could translate into enhanced fertility or reproductive success in the context of artificial insemination or natural mating.
The manuscript concludes with a statement on future research under in vivo conditions. It would be beneficial in the discussion to briefly elaborate on the potential implications or next steps for such research.
The "Conclusions" section is succinct and effectively summarizes the key findings of the study. The authors have appropriately highlighted the positive effects of fumaric acid at the concentration of 2.15mM on various aspects of post-thaw buck spermatozoa quality, namely viability, plasma and acrosome membrane integrity, and mitochondrial function.
The authors have also acknowledged the need for further research, particularly concerning the in vivo fertility of frozen-thawed buck spermatozoa treated with fumaric acid. This is an essential aspect to investigate, as the ultimate goal of such studies is to improve fertility outcomes in real-life reproductive scenarios. Additionally, the authors have identified the importance of understanding the mechanism of action of fumaric acid in spermatozoa, which would provide valuable insights into the underlying processes responsible for its beneficial effects.
The authors have included 106 references, a wide range of studies related to cryopreservation, sperm quality, oxidative stress, and antioxidant interventions. Having a substantial number of references can be beneficial. However, it is important to ensure that the references cited are relevant and add value to the article.
While having a substantial number of references can be informative, it is crucial to ensure that each reference contributes significantly to the article's content and does not lead to redundancy. The authors should carefully review the references and consider eliminating those that do not add substantial value or are very similar to other cited studies. By doing so, they can improve the overall clarity and focus of the article.
The manuscript needs some edits:
Lines 25-27: Different types of letters are observed.
In different parts of the manuscript, the "p" (p>0.05) is observed without italics, for example, lines 192, 214, 236, 246.
Reviewer 2 Report
The subject is exciting and increases knowledge on buck semen quality after soy lecithin extender supplemented with fumaric acid. The paper may contribute to establishing new strategies to enhance post-thaw spermatozoa quality. Unfortunately, there is no information available on the sperm fertilization capacity. This lack of information reduces the quality of the manuscript, even though the authors have mentioned that experiments are ongoing. Nonetheless, Veterinary Sciences should still consider publishing this manuscript. However, I have some suggestions that are necessary for improving the manuscript:
- It may be worth noting that many bibliographic citations in both the introduction and discussion are included, and the target species being discussed is not always clearly indicated in the text. One possible solution to this issue could be to limit the number of citations used and leave only the most recent and pertinent ones.
- The authors rightly state throughout the text that sperm damage is the result of excessive intracellular formation of reactive oxygen species (ROS), and also partially discuss the results in light of oxidative damage; however, they do not apply any test to verify the redox balance of their samples. If the main effect of FA is to serve as an antioxidant, further tests evaluating oxidative stress should be considered.
- The origin of the fumaric acid must be reported –magnesium fumarate?- (supplier company, code).
- In the results, when statistical differences are below 1%, please indicate p<0.01 (CASA motility data). Furthermore, as regards the results concerning sperm viability, plasma membrane functional integrity, acrosome integrity and mitochondrial membrane function, there is something wrong with the indication of the statistical significance. Please, pay special attention.
Reviewer 3 Report
This scientific manuscript deals with the effected of fumaric acid. It will be good if other yolk extenders were used and see its overall effect. it is a very simple and comprehensive manuscript. Its the starting point of using fumaric acid in a specific buck breed. I think that it is somewhat speculative because fumaric acid mechanism and in vivo effects were not evaluated. I think that a better description o some cinetic parameters would enrich this manuscript, if possible. This manuscripts deals more with men and boar. I think that its use in small ruminant would be good. Scale and numbers in figures are tool little. The effect of NRF2, is confuse. A better explanation is required.
Round 2
Reviewer 1 Report
We would like to thank the authors for the revisions made to the manuscript based on the comments received. The improvements incorporated have increased the clarity and completeness of the article.
The clarifications made in the Simple Abstract regarding the influence of oxidative stress on sperm structural and functional changes during cryopreservation has been effectively addressed.
The improved context provided in both the simple summary and the abstract, which clarifies the importance of improving sperm quality after thawing and the potential applications of the results and to highlight the practical implications of their research.
I would like to commend the inclusion of the total number of replicates or experiments performed for each treatment group in the Materials and Methods section. This addition provides readers with a better understanding of the robustness and statistical validity of the study.
The reduction of the bibliographic references by 25% is a great effort that will allow other researchers to focus on the most relevant references.